# Prospective Comparative Study of Etoposide plus G-CSF versus G-CSF Alone, Followed by Risk-Adapted Plerixafor for Peripheral Blood Stem Cell Mobilization in Patients with Newly Diagnosed Multiple Myeloma: CAtholic REsearch Network for Multiple Myeloma Study (CAREMM-2001)

**DOI:** 10.3390/cancers15194783

**Published:** 2023-09-28

**Authors:** Sung-Soo Park, Seung-Hwan Shin, Jung-Yeon Lee, Young-Woo Jeon, Seung-Ah Yhang, Chang-Ki Min

**Affiliations:** 1Hematology Hospital, Seoul St. Mary’s Hospital, The Catholic University of Korea, Seoul 02706, Republic of Korea; sspark@catholic.ac.kr (S.-S.P.); jungyeon1029@catholic.ac.kr (J.-Y.L.); ckmin@catholic.ac.kr (C.-K.M.); 2Myeoma Center, Hematology Institute, Eunpyeong St. Mary’s Hospital, The Catholic University of Korea, Seoul 03312, Republic of Korea; 3Department of Hematology, Yeouido St. Mary’s Hospital, The Catholic University of Korea, Seoul 07345, Republic of Korea; native47@catholic.ac.kr; 4Department of Hematology, Incheon St. Mary’s Hospital, The Catholic University of Korea, Incheon 21431, Republic of Korea; saymd@catholic.ac.kr

**Keywords:** etoposide, plerixafor, mobilization, multiple myeloma

## Abstract

**Simple Summary:**

We conducted a prospective trial comparing single-dose etoposide (375 mg/m^2^ for one day) plus G-CSF versus G-CSF alone, followed by risk-adapted plerixafor in myeloma patients. Despite significantly less frequent (*p* = 0.045) use of plerixafor in the etoposide group, the optimal collection rates (CD34+ cells ≥ 6 × 10^6^/kg) were not significantly different between the two groups (*p* = 0.195). In addition, the rate of collected CD34+ cells of ≥ 8.0 × 10^6^/kg was significantly higher in the etoposide group. Our results suggest that single-dose etoposide plus G-CSF may be a better option for patients who are expected to receive two or more transplantations.

**Abstract:**

To explore the optimal mobilization for multiple myeloma (MM) patients, we conducted a prospective trial comparing single-dose etoposide (375 mg/m^2^ for one day) plus G-CSF versus G-CSF alone, followed by risk-adapted plerixafor. After randomization, 27 patients in the etoposide group and 29 patients in the G-CSF alone group received mobilizations. Six (22.2%) patients in the etoposide group and 15 (51.7%) patients in the G-CSF alone group received plerixafor based on a peripheral blood CD34+ cell count of < 15/mm^3^ (*p* = 0.045). The median count of CD34+ cells collected was significantly higher in the etoposide group (9.5 × 10^6^/kg vs. 7.9 × 10^6^/kg; *p* = 0.018), but the optimal collection rate (CD34+ cells ≥ 6 × 10^6^/kg) was not significantly different between the two groups (96.3% vs. 82.8%; *p* = 0.195). The rate of CD34+ cells collected of ≥ 8.0 × 10^6^/kg was significantly higher in the etoposide group (77.8% vs. 44.8%; *p* = 0.025). Although the rates of grade II–IV thrombocytopenia (63.0% vs. 31.0%; *p* = 0.031) and grade I–IV nausea (14.8% vs. 0%; *p* = 0.048) were significantly higher in the etoposide group, the rates of adverse events were low in both groups, with no neutropenic fever or septic shock. Thus, both single-dose etoposide plus G-CSF and G-CSF alone with risk-adapted plerixafor were effective and safe, but the former may be the better option for patients who are expected to receive two or more transplantations.

## 1. Introduction

Even in the era of novel agents that have improved outcomes for patients with multiple myeloma (MM) over the past a few decades, disease relapse with subsequent therapeutic challenges remains inevitable [1]. As the disease-free and treatment-free intervals get progressively shorter as patients with MM receive later lines of therapy, first-line therapy is of paramount importance for their long-term survival and quality of life [2,3]. Therefore, novel agent-based induction followed by autologous stem cell transplantation (ASCT) is the preferred option for transplant-eligible patients with newly diagnosed MM (NDMM), as it provides more prolonged survival outcomes than chemotherapy alone [4,5,6]. Successfully performing ASCT for patients with NDMM without morbidity and mortality requires prerequisites, including acceptable tumor reduction by induction and patients’ tolerable condition [7]. Collection of CD34+ stem cells above thresholds is also critical to ensure sustained multilineage recovery and improve long-term post-transplant outcomes. Although there is considerable heterogeneity, most clinicians agree that CD34+ stem cells of 2 × 10^6^/kg or greater are required to avoid delayed or failed engraftment [8,9]. Meanwhile, others suggest that CD34+ stem cells of 5 × 10^6^/kg or greater are optimal for more rapid platelet recovery [10].

Until recently, CD34+ stem cell collection has been performed with steady-state mobilization of G-CSF alone and chemomobilization consisting primarily of chemotherapeutic agents such as intermediate-dose cyclophosphamide (Cy; 1.5–3.0 g/m^2^) or an intermediate-dose etoposide (375 mg/m^2^ for two days) plus G-CSF. However, these are occasionally associated with failure to collect adequate doses of CD34+ stem cells or significant toxicities, including infectious complications due to prolonged neutropenia [11]. This unmet need for more effective and safer mobilization strategies had not been addressed. Recently, the pivotal trial of plerixafor in patients with MM showed that it could improve the results of CD34+ stem cell collection without additional safety concerns [12], but the widespread use has been limited due to its high cost. To address this issue, Costa et al. [13] implemented the algorithm of G-CSF plus risk-adapted use of plerixafor, improving its cost-effectiveness. In addition, Park et al. [14] showed that single-dose etoposide (375 mg/m^2^ for one day) plus G-CSF could be an effective mobilization with a favorable safety profile.

Based on these studies, we conducted a prospective trial comparing the efficacy and safety of single-dose etoposide plus G-CSF (the etoposide group) versus G-CSF alone (the G-CSF alone group), followed by risk-adapted use of plerixafor. The results of the study are presented in this manuscript.

## 2. Patients and Methods

### 2.1. Study Design, Participants, and Randomization

This study is an open-label, prospective, comparative, and randomized trial in which patients were recruited from two tertiary referral hospitals of the Catholic University of Korea, Seoul, South Korea (Seoul St. Mary’s Hospital and Eunpyeong St. Mary’s Hospital, Seoul). Eligible patients were 18 years of age or older with histologically confirmed MM who achieved a partial response (PR) or better after the completion of four cycles of induction with proteasome inhibitors and/or immuno-modulatory agents. Participants were required to meet the following inclusion criteria: Eastern Cooperative Oncology Group (ECOG) performance status (PS) of two or less, adequate hematologic function (neutrophil count of 1.0 × 10^9^/kg or greater and platelet count of 75 × 10^9^/kg or greater), and normal hepatic (total bilirubin and aspartate/alanine aminotransferase less than two half times of the upper limit of normal) and renal (estimated glomerular filtration rate of 30 mL/min/1.73m^2^ or greater) function. Patients were excluded if they were ineligible for transplantation, had a history of previous autologous or allogeneic stem cell transplantation for other diseases, or had relapsed after first-line therapy without the intention of transplantation. Based on a preliminary analysis of our historical cohort, we estimated that 24 participants would be needed in each group to have approximately 99% discriminative power at the 5% significance level to detect the hypothesis that the etoposide group would have a significantly higher optimal collection rate. Considering the dropout rate of 20%, we planned to enroll 30 patients in each group. Using computerized random assignment, participants were randomly assigned 1:1 to the etoposide group and the G-CSF alone group. This trial was registered in the Clinical Research Information System of South Korea (KCT0006245).

### 2.2. Procedures

After randomization, participants in the etoposide group received a single-dose etoposide (375 mg/m^2^ intravenously [IV] for one day). G-CSF (lenograstim; 10 µg/kg subcutaneously [SC]) was administered to patients in the etoposide group (starting on the ninth day after the single-dose etoposide) and the G-CSF alone group until the last day of apheresis. On day five of G-CSF injection, we measured a PB CD34+ cell counts by flow cytometry. Patients with PB CD34+ cells less than 15/mm^3^ received an additional risk-adapted plerixafor (0.24 mg/kg SC) until completion of apheresis. CD34+ stem cell collection was performed in all patients using the COBE^®^ spectra (Terumo BCT, Lakewood, CO, USA) on day five of G-CSF injection, with one-day delay in those determined to receive a risk-adapted plerixafor, for a maximum of three days, with a target CD34+ stem cell count of 6 × 10^6^/kg (Figure 1). The number of bags in which the collected CD34+ stem cells were stored was determined at the discretion of the treating physician, with the goal of obtaining two or more storage bags containing at least 3 × 10^6^/kg of CD34+ stem cells. All patients who successfully collected at least 2 × 10^6^/kg of CD34+ stem cells proceeded to receive ASCT using a melphalan-based conditioning. To facilitate neutrophil engraftment, they received G-CSF (lenograstim; 10 µg/kg IV) from day five of CD34+ stem cell infusion to achieve a sustained absolute neutrophil count of 0.5 × 10^9^/L or greater. Other transplant-related procedures were performed according to each institution’s protocol.

### 2.3. Definition

Optimal and adequate collection were defined as achieving a CD34+ stem cell count of 6 × 10^6^/kg or greater and 3 × 10^6^/kg or greater, respectively, by the last day of apheresis. If the count of collected CD34+ stem cells was less than 2 × 10^6^/kg at the completion of apheresis, it was considered a collection failure. Pre- and post-mobilization responses to induction were assessed within two weeks after the completion of induction (before randomization) and after the completion of CD34+ stem cell collection (before conditioning), respectively, according to the IMWG response criteria [15], using serum and 24 h urine protein electrophoresis/immunofixation or serum-free light chain assay, with bone marrow aspiration and biopsy if patients were expected to achieve a complete response (CR) or better.

Neutrophil and platelet engraftment were defined as achieving a sustained neutrophil count of 0.5 × 10^9^/L or greater for at least three days and a sustained platelet count of 20 × 10^9^/L or greater for at least seven days, respectively. Adverse events were assessed using the National Cancer Institute Common Terminology Criteria for Adverse Events version 4.03 from mobilization to the completion of apheresis.

### 2.4. Endpoints and Statistical Analysis

The primary endpoint of the current study was the optimal collection rate of the etoposide group and the G-CSF alone group. The secondary endpoints were adequate collection rate, collection failure rate, cumulative incidence of neutrophil and platelet engraftment, and grade I–IV (non-hematologic toxicities) or grade II–IV (hematologic toxicities) or higher adverse events in both groups. Categorical variables in baseline and disease-related characteristics were described as counts with relative frequencies, whereas continuous variables were summarized as medians with ranges. All non-time-dependent endpoints were compared using the appropriate chi-squared or Fisher’s exact test. Neutrophil and platelet engraftment were estimated as a cumulative incidence, considering disease progression and death from any cause as competing risks, and compared using Grey’s test. Factors or covariates with *p* < 0.050 were considered significant using two-tailed significance testing. All statistical analyses were performed with R version 4.2.2 (R Foundation for Statistical Computing, Vienna, Austria) using the ‘Survival’, ‘tidycmprsk’, ‘ggplot2′, and ‘ggsurvfit’ packages on 15 May 2023.

## 3. Results

### 3.1. Baseline and Clinical Characteristics

Between 15 September 2021 and 31 October 2022, 60 NDMM patients who achieved PR or better after induction were recruited. Thirty patients were randomly assigned to the etoposide group and the others to the G-CSF alone group, respectively. Four patients (three in the etoposide group and one in the G-CSF alone group) withdrew informed consent prior to mobilization. The median age of all participants at randomization was 60 years (range, 37–69). According to the Reversed International Staging System [16], 17 (30.4%) and 27 (48.2%) patients had stages I and II–III disease, respectively. Prior to enrollment in the current study, patients received one course of induction, including four cycles of bortezomib plus thalidomide and dexamethasone (VTD) in 29 (51.8%), bortezomib plus lenalidomide and dexamethasone (VRD) in 4 (7.1%), and daratumumab plus VTD in 23 (41.1%). Pre-mobilization response to induction was CR or better in 21 (37.5%) patients, very good partial response (VGPR) in 23 (41.1%) patients, and PR in 12 (21.4%) patients. All baseline and disease-related characteristics were well balanced between the etoposide and the G-CSF alone groups (*p* ≥ 0.199) (Table 1).

### 3.2. Mobilization

The median PB CD34+ cell count on day five of G-CSF injection was significantly higher in the etoposide group (29.8/mm^3^ vs. 15.5/mm^3^; *p* = 0.001). Based on a measured count of less than 15/mm^3^, six (22.2%) patients in the etoposide group and 15 (51.7%) patients in the G-CSF alone group received risk-adapted plerixafor (*p* = 0.045) (Figure 2A). In addition, the median duration of apheresis was significantly shorter in the etoposide group than in the G-CSF alone group (1 day vs. 2 days; *p* = 0.027). Consequently, the median count of CD34+ stem cells collected was significantly higher in the etoposide group (9.5 × 10^6^/kg vs. 7.9 × 10^6^/kg; *p* = 0.018) (Figure 2A). However, the optimal collection rate was not significantly different between the etoposide group and the G-CSF alone group (96.3% vs. 82.8%; *p* = 0.195) (Figure 2B). When we performed the post hoc analysis, the rate of collected CD34+ stem cells of 8.0 × 10^6^/kg or greater was significantly higher in the etoposide group (77.8% vs. 44.8%; *p* = 0.025) (Figure 2B). The optimal collection rate on apheresis day one was significantly higher in the etoposide group (77.8% vs. 44.8%; *p* = 0.025), but did not differ significantly differ between the two groups on apheresis day two (96.3% vs. 82.8%; *p* = 0.195) (Figure 2C). All patients in both groups achieved collected CD34+ stem cell counts of 3.0 × 10^6^/kg or greater. The collected CD34+ stem cells were stored in a median of two (range: 2–4) storage bags in the etoposide group and two (range: 1–4) storage bags in the G-CSF alone group (*p* = 0.075). In patients who achieved optimal collection, the median count of CD34+ stem cells remaining after one course of ASCT was 4.8 × 10^6^/kg (range, 2.8 × 10^6^/kg–6.7 × 10^6^/kg). On the other hand, no patient in either group showed an improved response after mobilization followed by CD34+ stem cell collection compared to before mobilization. Baseline and disease-related characteristics did not significantly affect the rates of optimal collection and using risk-adapted plerixafor (*p* ≥ 0.092).

### 3.3. Adverse Events

The rates of adverse events observed in both the etoposide and the G-CSF alone groups were relatively low, not exceeding 20.0%, except for grade II–IV thrombocytopenia of 46.4% (95% CI, 33.0–60.3). All were self-limited or manageable and resolved within three days. There were no infectious complications such as neutropenic fever or septic shock. No serious adverse events requiring hospitalization or resulting in permanent disability or death were observed. When comparing the rates of hematologic and non-hematologic adverse events between the etoposide and the G-CSF alone groups, the rates of grade II–IV thrombocytopenia (63.0% vs. 31.0%; *p* = 0.031) and grade I–IV nausea (14.8% vs. 0%; *p* = 0.048) were significantly higher in the etoposide group. However, there was no significant difference in the rate of grade III–IV thrombocytopenia rate between the etoposide and G-CSF alone groups (14.8% vs. 17.2%; *p* = 1.000). No clinical event of bleeding or hemorrhage was observed in either group. In addition, transient grade II–IV leukopenia (3.7% vs. 0%; *p* = 0.482) and neutropenia (7.4% vs. 3.4%; *p* = 0.605) were observed in only a small number of patients in both groups. There were no significant differences in the rates of other adverse events between the two groups as described in Table 2.

### 3.4. Neutrophil and Platelet Engraftment

All patients in the etoposide and the G-CSF alone groups successfully proceeded to ASCT. The conditioning regimens were melphalan at 100 mg/m^2^ IV for two days in 53 (94.6%) patients and melphalan at 70 mg/m^2^ IV for two days plus busulfan at 3.2 mg/kg IV for three days in 3 (5.4%) patients. Of those receiving melphalan conditioning, the dose was reduced to 70 mg/m^2^ in 11 patients (20.6%) due to an estimated glomerular filtration rate of less than 40 mL/min/1.73m^2^ or other comorbidities. The median count of infused CD34+ stem cells was not significantly different between the etoposide group and the G-CSF alone group (5.8 × 10^6^/kg vs. 4.9 × 10^6^/kg; *p* = 0.162). One patient in the etoposide group experienced delayed platelet engraftment (on day 211 after transplantation), but the median days to neutrophil engraftment (11 days in both; *p* = 1.000) and platelet engraftment (11 days in both; *p* = 1.000) were not significantly different between the two groups (Figure 3). All patients who survived to the last day of follow-up maintained neutrophil and platelet engraftment.

## 4. Discussion

The current randomized trial comparing the mobilizations with single-dose etoposide plus G-CSF and G-CSF alone, followed by risk-adapted plerixafor showed that optimal collection rates were not significantly different between the two groups. However, the rate of collected CD34+ stem cells of 8.0 × 10^6^/kg or greater was significantly higher in the etoposide group. Patients in the etoposide group also required significantly fewer days of apheresis. The incidence of adverse events was acceptable in both groups. These findings suggest the considerations for the optimal mobilization strategy in NDMM patients who are expected to receive an ASCT.

Steady-state mobilization with G-CSF alone and chemomobilization mainly with intermediate-high-dose cyclophosphamide (1.5–3.0 g/m^2^) plus G-CSF have been the two mainstays for mobilizing CD34+ stem cells from bone marrow. However, when we used these traditional strategies, some challenges to collecting optimal doses of CD34+ stem cells were encountered, such as the following: mobilization with G-CSF alone was associated with a high collection failure rate of 24–50%, despite its significantly lower toxicity rate [17,18]. While an intermediate-high-dose Cy-based mobilization improved the successful collection rates up to 90%, it was associated with occasionally fatal toxicities, including neutropenic fever and septic shock up to 20% [19,20,21].

On the armamentarium to overcome these challenges, etoposide (topoisomerase II inhibitor) has been a widely used chemotherapeutic agent to facilitate the collection of CD34+ stem cells in patients with various hematologic diseases. Although the detailed mechanism to mobilize hematopoietic stem cells (HSCs) from bone marrow needs to be further elucidated, Kang et al. recently showed that the etoposide increased the interleukin-8 secretion by bone marrow stromal cells, which is responsible for the enhanced proliferation and mobilization of HSCs [22]. In contrast, its clinical role in CD34+ stem cell mobilization has been investigated in several previous studies. Wood et al. [23] reported that the results of CD34+ stem cell collection using intermediate-dose etoposide (375 mg/m^2^ for two days) plus G-CSF mobilization in 152 patients with MM, which showed that 150 (98.7%) patients could achieve optimal collection (CD34+ cells of 5 × 10^6^/kg or greater for one to two days), but the toxicities of neutropenia (78.9%) and neutropenic fever (9.0%) were of concern to be overcome. Subsequently, Park et al. reported the results of CD34+ cell collection in 32 patients with MM using single-dose etoposide (375 mg/m^2^ for one day) plus G-CSF mobilization, which showed a better safety profile of febrile neutropenia in two (6.3%) patients, without other toxicities such as mucositis, pneumonia, and bacteremia, and without decreased rates of optimal (CD34+ cells of 5 × 10^6^/kg or greater; 71.9%) and adequate (CD34+ cells of 3 × 10^6^/kg or greater; 87.5%) collection.

Plerixafor is a selective and reversible antagonist of CXCR4, which induces chemotaxis in the bone marrow through its ligand of SCF-1. In its pivotal phase III trial in 302 patients with MM, the plerixafor group showed a higher rate of successful collection (CD34+ cells of 6 × 10^6^/kg or greater) compared to the placebo group (71.6% vs. 34.4%; *p* < 0.001) [12]. The most common adverse events with plerixafor were mild gastrointestinal toxicity (diarrhea in 18.4%, nausea in 16.3%, and vomiting in 5.4%) and injection site erythema (20.4%). However, the high cost of plerixafor has limited its widespread use in real-world practice. In light of this, Costa et al. [24] implemented a decision algorithm using the PB CD34+ cell count on day five of G-CSF injection to optimize the cost-effectiveness using risk-adapted plerixafor. When they evaluated it in a validation cohort of 34 patients with MM, 33 (97.1%) achieved the pre-specified CD34+ stem cell targets (6 × 10^6^/kg for patients expected to receive tandem ASCT and 3 × 10^6^/kg for others). Given the high applicability of this approach in real-world practice, we used it in both the etoposide and the G-CSF alone groups of the current study, using a PB CD34+ cell threshold of 15/mm^3^ (the highest to optimize the burden by using plerixafor, for a target CD34+ cell collection of 3 × 10^6^/kg, in the study by Costa et al. [24]).

The results of the current study did not meet the hypothesis for the primary endpoint that the optimal collection rate would be significantly higher in the etoposide group. However, the PB CD34+ cell count on day five of G-CSF injection was significantly higher in the etoposide group, suggesting that the optimal collection rates of the two groups could be significantly different if patients did not receive a risk-adapted plerixafor. That is, a stricter protocol-driven use of risk-adapted plerixafor in a prospective setting compared to the historical real-world cohort, based on the sample size calculation in the current study, might make the optimal collection rates between the two groups not significantly different. In addition, we defined the optimal collection as the collected CD34+ stem cells of 6 × 10^6^/kg according to the policy of our institutions, which is based on a previous report using G-CSF alone, followed by risk-adapted plerixafor [24]. However, the optimal threshold for CD34+ stem cell collection has not yet been determined. Considering that the CD34+ stem cells of 2 × 10^6^/kg or greater and of 5 × 10^6^/kg or greater, respectively, are required to avoid delayed or failed engraftment and to achieve more rapid platelet recovery [8,9,10], it is possible to choose a reasonable count between CD34+ stem cells of 4 × 10^6^/kg to 10 × 10^6^/kg or greater to be determined in MM patients with the possibility of receiving a second course of ASCT, as in previous reports and guidelines [14,23,25]. Therefore, the conclusion of the current study should be interpreted with caution, considering that it may be modified according to the definition of optimal collection.

Considering the results of the current study, it is suggested that both single-dose etoposide plus G-CSF and G-CSF alone, followed by risk-adapted plerixafor are acceptable mobilizations in patients with MM who are expected to receive at least one course of ASCT. In addition, they are consistent with those of recent pivotal Phase III trials using modern triplet and quadruplet induction. Patients in the CASSIOPEIA trial achieved a mean count of CD34+ stem cells collected of 6.3 × 10^6^/kg in the daratumumab plus VTD group and 8.9 × 10^6^/kg in the VTD group [26]. However, it should be noted that these were achieved with an intermediate-dose Cy (2.0–3.0 g/m^2^) mobilization. In addition, patients in the GRIFFIN trial achieved a median count of CD34+ stem cells collected of 8.3 × 10^6^/kg in the daratumumab plus VRD group and 9.4 × 10^6^/kg in the VRD group with G-CSF alone mobilization. However, the proportion of patients receiving upfront or rescue plerixafor (69.5% in the daratumumab + VRD group and 56.3% in the VRD group) was considerably higher compared to our cohort [27].

However, the current study suggests that the single-dose etoposide plus G-CSF followed by risk-adapted plerixafor may be more helpful than the G-CSF alone in real-world practice. First, single-dose etoposide plus G-CSF may make it possible to reduce the burden of CD34+ stem cell collection by suppressing the use of host-cost plerixafor, which is supported by previous studies showing that the optimal use of plerixafor contributes to reducing the cost of the CD34+ stem cell collection [24,25]. In addition, etoposide may not require a high burden to manage toxicities due to their low severity and incidence. Second, considering that the rate of collected CD34+ stem cells of 8.0 × 10^6^/kg or greater was significantly higher in the etoposide group, single-dose etoposide plus G-CSF may be a better choice for patients with specific disease-related conditions, including patients with extra-medullary disease and high-risk cytogenetics who require a tandem ASCT [28,29]. It should also be noted that salvage ASCT showed better results than chemotherapy alone in relapsed/refractory MM patients with a relatively longer relapse-free interval after the first course of ASCT [30,31,32]. Furthermore, concerns have been raised that CD34+ stem cells may be impaired by daratumumab and/or lenalidomide-containing induction [20,33,34]. In a retrospective study of patients receiving daratumumab + VTD induction, Liberatore et al. [35] showed that the inhibitory effect of daratumumab on mobilization could be overcome by the use of high-dose cyclophosphamide (4.0 g/m^2^), with a mean count of CD34+ stem cells collected of 10.68 × 10^6^/kg (range, 4.92–18.8). Considering the significantly higher count of CD34+ stem cells collected in our etoposide group, it is reasonable to suggest that a single-dose etoposide may be helpful in patients receiving daratumumab and/or lenalidomide-containing induction. However, the current study could not address this issue due to the relatively small number of participants, which should be explored in further studies with large numbers of participants.

## 5. Conclusions

In the current study, optimal collection rates were acceptable for patients receiving single-dose etoposide plus G-CSF and G-CSF alone, followed by risk-adapted plerixafor. However, the rate of collected CD34+ stem cells of 8.0 × 10^6^/kg or greater was significantly higher in the etoposide group. Meanwhile, the rates of adverse events were relatively low in both groups. These results suggest that single-dose etoposide plus G-CSF may be a better choice, at least for patients expected to receive a second or third course of ASCT. The limitations of the current study, including the relatively small number of participants, which made it difficult to perform a detailed analysis of factors influencing the outcome of CD34+ stem cell collection, could be overcome by well-designed studies with larger numbers of patients to provide sufficient power to address these issues.

## Figures and Tables

**Figure 1 cancers-15-04783-f001:**
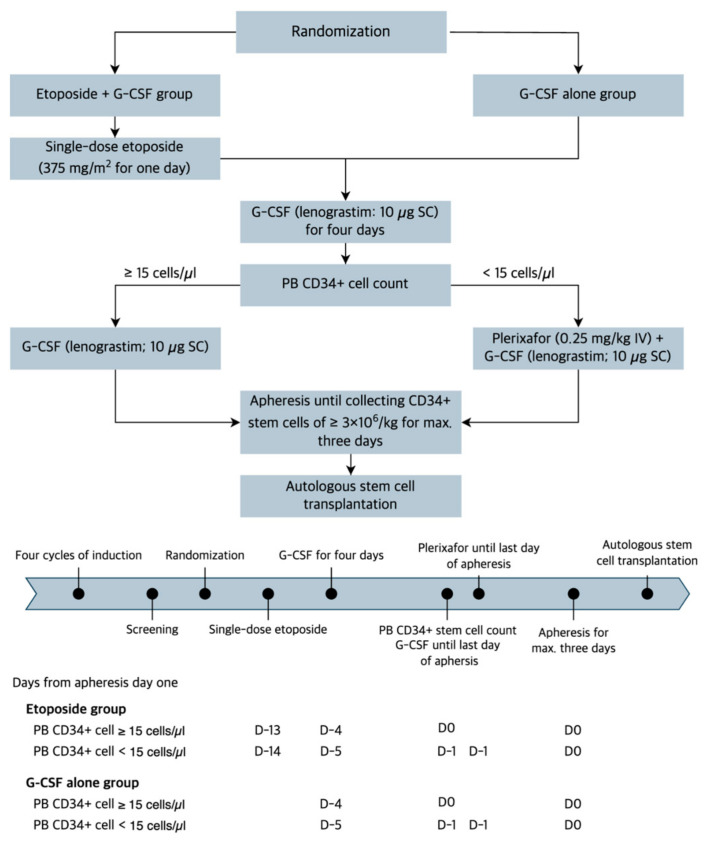
Trial scheme.

**Figure 2 cancers-15-04783-f002:**
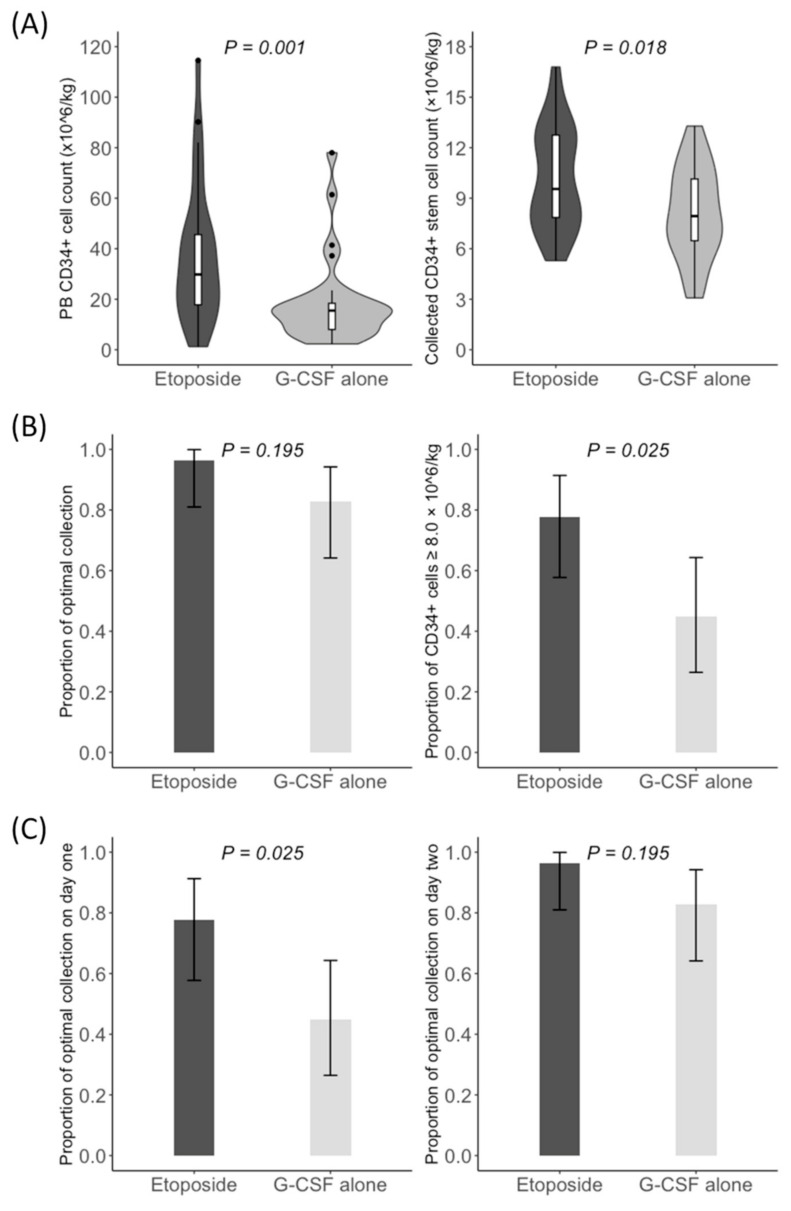
Mobilization outcomes. (**A**) The counts of PB CD34+ cells on day five of G-CSF injection and collected CD34+ stem cells; (**B**) rates of optimal collection and collected CD34+ stem cells of 8.0 × 10^6^/kg or grater; (**C**) rates of optimal collection on day one and two of apheresis.

**Figure 3 cancers-15-04783-f003:**
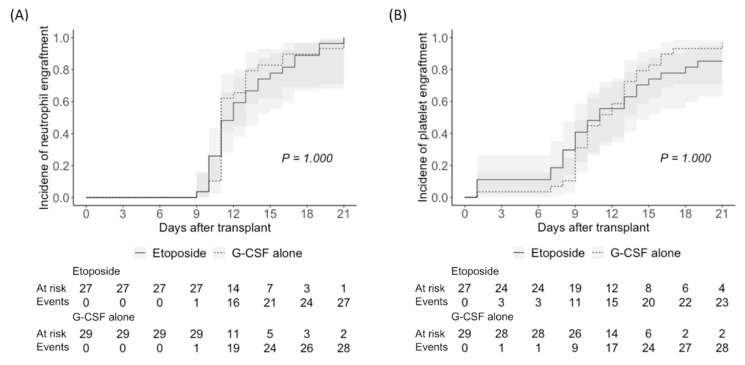
(**A**) Neutrophil and (**B**) platelet engraftment after autologous stem cell transplantation.

**Table 1 cancers-15-04783-t001:** Baseline clinical and disease-related characteristics of the etoposide + G-CSF and G-CSF alone groups.

Characteristics	Etoposide + G-CSF Group	G-CSF Alone Group	*p*
Number of patients	27	29	
Age, median (range)	58 (49–68)	60 (37–69)	0.611
65 yrs/>65 yrs	22 (81.5%)/5 (18.5%)	22 (75.95)/7 (24.1%)	0.748
Sex			
Male/female	14 (51.9%)/13 (48.1%)	16 (55.2%)/13 (44.8%)	1.000
ECOG PS			
<2/≥2	15 (71.4%)/6 (28.6%)	16 (76.2%)/5 (23.8%)	1.000
Myeloma isotype			0.237
Ig G/Ig A	17 (63.0%)/5 (18.5%)	18 (62.1%)/2 (6.9%)	
Light chain disease	4 (14.8%)	9 (31.0%)	
Others (Ig D)	1 (3.7%)	0	
Serum albumin			
<3.5 g/dL/≥3.5 g/dL	19 (70.4%)/8 (29.6%)	19 (67.9%)/9 (32.1%)	1.000
β_2_ microglobulin			
<5.5 mg/L/≥5.5 mg/L	14 (53.8%)/12 (46.2%)	18 (66.7%)/9 (33.3%)	0.501
LDH			
≤ULN vs. >ULN	19 (70.4%)/8 (29.6%)	25 (86.2%)/4 (13.8%)	0.199
eGFR (ml/min/1.73m^2^)			
≥60/<60 mL/min	21 (77.8%)/6 (22.2%)	25 (89.3%)/3 (10.7%)	0.295
Cytogenetic risk by FISH			
Standard vs. high	20 (87.0%)/3 (13.0%)	22 (88.0%)/3 (12.0%)	1.000
Disease stage by R-ISS			
I/II–III	9 (36.0%)/16 (64.0%)	8 (29.6%)/19 (70.4%)	0.847
Induction regimen			1.000
VTD/VRD	14 (51.9%)/2 (7.4%)	15 (51.7%)/2 (6.9%)	
Dara + VTD	11 (40.7%)	12 (41.4%)	
Pre-mobilization response			0.485
CR/VGPR	12 (44.4%)/9 (33.3%)	9 (31.0%)/14 (48.3%)	
PR	6 (22.2%)	6 (20.7%)	

**Table 2 cancers-15-04783-t002:** Hematologic and non-hematologic adverse events of the etoposide + G-CSF and G-CSF alone groups.

Adverse Events	Etoposide + G-CSF Group	G-CSF Alone Group	*p*
**Hematologic toxicities**			
Leukopenia (grade II–IV))	1 (3.7%)	0	0.482
Neutropenia (grade II–IV)	2 (7.4%)	1 (3.4%)	0.605
Thrombocytopenia of			
grade II–IV	17 (63.0%)	9 (31.0%)	0.031
grade III–IV	4 (14.8%)	5 (17.2%)	1.000
Anemia (grade II–IV)	9 (33.3%)	4 (13.8%)	0.116
**Non-hematologic toxicities (grade I–IV)**			
GI disorders			
Nausea	4 (14.8%)	0	0.048
Vomiting	1 (3.7%)	0	0.482
Diarrhea	0	1 (3.4%)	1.000
Musculoskeletal disorders			
Bone pain	4 (14.8%)	4 (13.8%)	1.000
Headache	1 (3.7%)	4 (13.8%)	0.353
Paresthesia	3 (11.1%)	3 (10.3%)	1.000
Others			
General weakness	0	1 (3.4%)	1.000
Catheter-associated pain	0	1 (3.4%)	1.000

## Data Availability

Data will be made available upon request to the corresponding author (SHS) due to restrictions of the Personal Information Protection Act of South Korea.

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
