# Peer review of "Prospective Comparative Study of Etoposide plus G-CSF versus G-CSF Alone, Followed by Risk-Adapted Plerixafor for Peripheral Blood Stem Cell Mobilization in Patients with Newly Diagnosed Multiple Myeloma: CAtholic REsearch Network for Multiple Myeloma Study (CAREMM-2001)"

_cancers, 2023, doi:10.3390/cancers15194783_

Round 1

Reviewer 1 Report

Park et al. report a prospective trial evaluating the use of etoposide for stem cell collection in multiple myeloma patients. While the study protocol is sound, the sample size is small and citations are missing. Furthermore, a decent discussion regarding the use of etoposide versus cyclophosphamide is missing.

MAJOR:

Line 155: "They received one course of induction". Was the stem cell collection performed after one or four cycles of induction? Please clarify

Please report the number of "stem cell bags", in order to see how often patients might be transplanted (1,2,3 times). Comment on the collection goal, was is three bags in every patient?

"Meanwhile, the rates of adverse events of were relatively low, with no neutropenic fever or septic shock observed in either group. A" - Rates of thrombocytopenia and nausea where significantly higher with etoposide, please acknowledge in abstract

Please report on the detailed response parameters both before and after stem cell collection (CR, VGPR, PR). Any improvent with etoposide therapy?

Please compare the stem cell yields with recent publications from the GRIFFIN and the CASSIOPEIA trials.

Important citations regarding recent publications on stem cell collection after triplet or quadruplet induction therapies are missing. Please add: PMID 36494017, 36200424, 37439346

MINOR:

Please use larger font size for figures.

Please add p values to figures

Please carefully check for typos, e.g.,  "grater"

The quality of English language needs moderate editing.

Reviewer 2 Report

The work by Park et al. presents a very well designed study with compelling results that are of clinical significance. The paper is overall very well written and the authors show that using Etoposide as an additional stem cell mobilizer significantly improves stem cell yield with no significant side effects.

One main concern is that the protocol procedure and administration of Etoposide could be described better. Figure 1 shows that one single dose of Etoposide is given on Day1 followed by G-CSF for 4 days and then by further G-CSF or Plerixafor afterwards. On what day do patients on the Etoposide arm collect stem cells after Etoposide administration and how does that compare to the G-CSF arm? Is there no transient neutropenia for patients taking Etoposide? The text mentions in line 104 that Etoposide was administered on the ninth day of G-CSF administration, that is not what the figure shows. Is that a mistake?

Further comments as follows

-          The word “optimal collection rate” is a term that appears to be based on institutional guidelines and should be explained as such. Eg. the current work uses a minimum of 6x10^6/kg CD34 cells as optimal rate, however the minimum goal of stem cells to collect can be 10x10^6 or even higher in other institutions, in which case there would be a significant difference between these arms. It is hence important that the authors define their optimal collection rate already in the “Simple Summary” (and also in the methods section and emphasize that this rate can vary by institution (in the discussion).

-          For figure 2 please add error bars and significance into the figures. It would also be ideal if all figures could either use bars (like in 2B and 2C) or violin plots (like in 2A)

Otherwise just some typos

-          Line 35- Meanwhile, the rates of adverse events of were relatively low…-> please delete “of”

-          Line 48- autologous stem cell transplantation followed by novel agent-based induction -> please change to “…. following novel agent-based induction”

-          Line 68- ….. has been limited due to its high burden-> do you mean high costs??

Round 2

Reviewer 1 Report

The authors have adequately responded to the suggestions .

.

English language seems to be appropriate.